# Zeta CrAss-like Phages, a Separate Phage Family Using a Variety of Adaptive Mechanisms to Persist in Their Hosts

**DOI:** 10.3390/ijms26167694

**Published:** 2025-08-08

**Authors:** Igor V. Babkin, Valeria A. Fedorets, Artem Y. Tikunov, Ivan K. Baykov, Elizaveta A. Panina, Nina V. Tikunova

**Affiliations:** Institute of Chemical Biology and Fundamental Medicine, Siberian Branch of the Russian Academy of Sciences, Novosibirsk 630090, Russia; f.valeriya41@gmail.com (V.A.F.); arttik@ngs.ru (A.Y.T.); baykov@niboch.nsc.ru (I.K.B.); e.panina@g.nsu.ru (E.A.P.)

**Keywords:** crAss-like phages, genome, virus taxonomy, stop codon recoding, phylogenetics, diversity-generating retroelements

## Abstract

Bacteriophages of the order Crassvirales are highly abundant and near-universal members of the human gut microbiome worldwide. Zeta crAss-like phages comprise a separate group in the order *Crassvirales,* and their genomes exhibit greater variability than genomes of crAss-like phages from other families within the order. Zeta crAss-like phages employ multiple adaptation mechanisms, ensuring their survival despite host defenses and environmental pressure. Some Zeta crAss-like phages use alternative genetic coding and exploit diversity-generating retroelements (DGRs). These features suggest complex evolutionary relationships with their bacterial hosts, sustaining parasitic coexistence. Mutations in tail fiber proteins introduced by DGR can contribute to their adaptation to changes in the host cell surface and even expand the range of their hosts. In addition, the exchange of DNA polymerases via recombination makes it possible to overcome the bacterial anti-phage protection directed at these enzymes. Zeta crAss-like phages continuously adapt due to genetic diversification, host interaction tweaks, and counter-defense innovations, driving an evolutionary arms race with hosts. Based on the genome characteristics of the Zeta crAss-like phages, we propose to separate them into the *Echekviridae* family (“эчәк”—“intestines” in Tatar) following the tradition of using the word “intestines” in different languages, suggested previously.

## 1. Introduction

CrAss-like phages are a group of bacteriophages that are related to the crAssphage (cross-assembly phage), a bacteriophage discovered through computational analysis of human gut metagenomic data in 2014 [1]. Advanced computer analysis of extended sequence databases has provided predictions of the function of most crAssphage genes [2,3,4]. Subsequent studies of related genes have indicated numerous crAss-like phages in various host-associated and environmental viromes, including human and animal guts, sewage, and marine ecosystems [5,6,7,8,9,10,11]. CrAss-like phages are a fascinating and highly prevalent group of gut bacteriophages with potential implications for human health and microbiome study. They primarily infect bacteria of the phylum *Bacteroidota*, which are dominant members of the human gut microbiota. Due to their high abundance, crAss-like phages likely play a significant role in modulating gut bacterial populations. They can influence nutrient cycling, bacterial evolution, and even human health by affecting the gut microbiome balance [6,12,13]. Some studies suggest crAss-like phages may be linked to obesity, inflammatory bowel disease, and other gut-related conditions [14,15,16,17]. Since they are widely present in human feces, it has been proposed to use crAss-like phages as markers for human fecal contamination in water sources [13,18,19,20,21,22].

CrAss-like phages were classified into multiple clades, namely, Alpha, Beta, Gamma, Delta, Zeta, and Epsilon based on genomic similarity [3,4]. They belong to the proposed viral families in the order *Crassvirales*. The International Committee on Taxonomy of Viruses (ICTV) has officially classified four clades of crAss-like phages into four distinct families: *Intestiviridae*, *Steigviridae*, *Crevaviridae*, and *Suoliviridae* [6,23]. These new taxonomic groups correspond to the previously designated Alpha, Beta, Gamma, and Delta clades, respectively, reflecting advancements in understanding their genetic diversity and evolutionary relationships. CrAss-like phages of clades Zeta and Epsilon remain the least studied.

CrAss-like phages are conserved across human populations, suggesting a long evolutionary relationship with humans; however, crAss-like phages exhibit notable variations in prevalence, genetic diversity, and host specificity across different human populations [6,14,24,25]. These differences are influenced by factors such as geography, diet, lifestyle, and host genetics, which shape the gut microbiome. For example, crAss-like phages from the *Intestiviridae* and *Suoliviridae* families are nearly ubiquitous in industrialized populations but show lower prevalence in traditional hunter-gatherer and rural communities. This disparity may reflect differences in diet and hygiene that affect *Bacteroides* host abundance [7,26,27,28].

CrAss-like phages possess unique genetic characteristics, most notably, a sophisticated transcriptional system featuring an atypically structured DNA-dependent RNA polymerase. This viral polymerase bears structural resemblance to eukaryotic RNA-dependent RNA polymerases, which are known to participate in RNA interference. Structural and functional studies have verified that this distinctive polymerase is a core component of the phage virion. During infection, it is delivered into the host cell together with the phage’s genetic material, where it initiates transcription of the phage’s early-stage genes [4,29,30].

A recent study has attempted to systematize current knowledge about the diversity and genomic organization of Epsilon crAss-like phages. It has been shown that these phages differ significantly from other crAss-like phages, which indicates the need to classify these phages into a separate family [31]. In this study, Zeta crAss-like phage genomes were analyzed. These phages have the largest genome in the order, roughly two times the average genome of four well-known phages from the *Intestiviridae*, *Steigviridae*, *Crevaviridae*, and *Suoliviridae* families, whose genomes vary from 42 kb to 149 kb; and most of the genomes are 70–80 kb. Zeta crAss-like phages encode a complex replication apparatus, supplemented by deoxyribonucleotide metabolism enzymes, and demonstrate significant heterogenicity of the sequences within the family. Zeta crAss-like phages clearly belong to the order *Crassvirales*, in which they form a monophyletic clade of viruses and share genes with other crAss-like phage families, such as the gene of viral RNA polymerase, which differs significantly from other RNA polymerase genes. At the same time, Zeta crAss-like phages have significant differences from other crAss-like virus families. Based on this study, we propose to name the Zeta crAss-like phage family *Echekviridae* that is a language derivative of the word “эчәк”—“intestines” in Tatar, aligning with the established practice of naming crAss-like phage families after intestinal terms in various languages, as was suggested previously [6].

## 2. Results

### 2.1. Analysis of the Lera Phage Genome

A virome shotgun library was constructed from a stool sample obtained from a healthy volunteer (Project: PRJNA1025976; BioSample: SAMN37731570; SRA: SRR26322353). De novo assembly using the SPAdes genome assembler V.3.15.2 revealed a 179,624 bp gapless contig with a pseudo-circular structure, suggesting that this is the complete genome of the unknown Caudoviricetes phage. The starting point of the genome was identified using PhageTerm analysis. This phage was named Lera, and its complete genome sequence was submitted into the GenBank database with the accession number PV725988.

After identification and annotation of the ORFs using the standard genetic code, it was found that this genome does not encode the full terminase large subunit (TerL) as the TGA stop codon destroys the TerL gene. In addition, the TGA stop codon disrupts the putative ORFs encoding DNA-dependent RNA polymerase and helicase. All mentioned genes were restored using the alternative code (Translation Table 4, https://www.ncbi.nlm.nih.gov/Taxonomy/Utils/wprintgc.cgi, accessed on 7 August 2025), in which TGA encodes tryptophan (Trp). Thus, Translation Table 4 was used to determine ORFs in the Lera genome.

The genome of the Lera phage encodes 257 putative ORFs including 26 tRNA genes (Appendix A). Of the remaining 231 ORFs, 111 encode proteins with predicted functions, which were determined based on the similarity of their amino acid sequences and domain structures with known phage proteins. The remaining 120 ORFs were signed as hypothetical proteins. Forty-one ORFs contain TGA-suppressed stop codons (Appendix A). Among them, 24 ORFs encode proteins with predicted functions, and most of them are structural proteins, including the muzzle protein and two of the three ring protein genes. Of the four putative lysine genes, three genes contain TGA codons, as does the holin gene. To confirm that the TGA codon corresponds to Trp in the genome of the Lera phage, an analysis of related sequences from GenBank was performed, which revealed that the “destroyed” proteins show clear similarities with other phage proteins with Trp at the suppression sites.

Of the twenty-six tRNA genes, two genes encode suppressor tRNAs in the Lera genome: tRNA-SUP-TCA (opal suppression) and tRNA-SUP-CTA (amber suppression). Prediction of the tRNA-SUP-TCA isotype (S1 data) using tRNAscan-SE version 2.0 showed that this tRNA corresponds to either Trp or arginine. It has been previously shown that in most phage genomes, tRNA-SUP-TCA corresponds to Trp [32,33]. Notably, no signs of amber suppression were found in the genome; all ORFs either do not use the TAG as a stop codon, or TAG is located at the end of the ORF and there is another stop-codon TAA immediately after TAG.

The Ori-Finder program predicted the position (1264-1624) of the replication origin in the Lera genome (Appendix A). Counting from the oriC region, two large blocks of genes were found in the genome. The first block of the genes oriented in the forward direction contains the genes of nucleic acid metabolism (including DNA polymerase), most tRNA genes, and four genes of proteins involved in tRNA metabolism, namely, CCA tRNA nucleotidyltransferase, tRNA pseudouridine synthase, aminoacyl-tRNA hydrolase, and aspartyl/glutamyl-tRNA (Asn/Gln) amidotransferase. In addition, five ORFs encoding homing endonucleases can be noted, which indicates the presence of introns/inteins in the genome of the Lera phage. The second block of the genes oriented in the reverse direction comprises the genes encoding virion and packaging proteins. In addition, a diversity generating retroelement (DGR-cassette) is included in the second block of genes.

### 2.2. Comparative Analysis of the Lera Phage Genome

To determine the clade that the Lera phage genome belongs to, its TerL protein sequence was compared with similar sequences from the GenBank database. As a result, BLASTp (https://blast.ncbi.nlm.nih.gov/Blast.cgi?PROGRAM=blastp&PAGE_TYPE=BlastSearch&LINK_LOC=blasthome, accessed on 19 May 2025) revealed 60 sequences with an evolutionary divergence of less than 1.4, belonging to phages that are marked as unclassified in GenBank (Figure 1). In addition, five relative TerL sequences (OMBT01000004, OLGM01000006, OMFN01000011, OBKV01000026, and OLVX01000015) were extracted from the previous study of the metagenome-assembled viral genomes from the human gut [4], and these proteins were defined as TerL of Zeta crAss-like phages in the study.

Phylogenetic analysis indicated that the Lera TerL protein sequence and 60 related sequences demonstrate high homology with five TerL sequences of Zeta crAss-like phages and form a separate monophyletic clade in a dendrogram containing 259 TerL sequences of other crAss-like phages including *Intestiviridae*, *Steigviridae*, *Crevaviridae*, *Suoliviridae*, and Epsilon crAss-like phages (Figure 1). This data suggests that the Lera phage and 60 undefined phages are, in fact, Zeta crAss-like phages.

To analyze Zeta crAss-like genomes, 42 sequences with a length of more than 150 kb were selected from the available data containing the TerL gene related to the Lera one. Previously, sequences of this length have been characterized as pseudocircular MAGs [4], indicating that these sequences were complete phage genomes. Therefore, only the complete or near-complete phage genomes were used for further analysis. The analysis of the proteomic tree calculated using the ViPTree web server confirmed the grouping of the Lera phage with Zeta crAss-like phages (Figure 2).

About half of the Zeta crAss-like genomes are ~180 kb or more in length. SG values were calculated according to [34] as normalized values of tBLASTx. Based on SG values, Zeta crAss-like phages are closest to Beta crAss-like phages (*Steigviridae*). Notably, Beta crAss-like genomes are ~100 kb in length. The significant differences between the Zeta crAss-like phage proteomes and other crAss-like ones make it possible to classify Zeta crAss-like phages into a separate family. A name for the proposed family is suggested to be *Echekviridae*, which is language derivative of the word эчәк—“intestines” in Tatar (local people in Siberia) in accordance with the tradition, introduced previously [6].

To clarify the taxonomy of Zeta crAss-like phages, an intergenomic similarity matrix was constructed using the viral intergenomic distance calculator (VIRIDIC) (Figure 3). VIRIDIC is commonly used to classify phages into species (95% similarity) and genera (70% similarity) [35,36]. A comparative analysis of the genome was performed based on intergenomic similarity, aligned genome share, and the ratio of genome lengths. The obtained results indicated a significant diversity of the genomes of this proposed family. There are seven putative genera within this family represented by two or more viruses.

The complete Lera genome was compared with eight phage genomes from various clades (Figure 2) of the proposed *Echekviridae* family. The starting points of these genomes were aligned with the Lera genome, and then signature genes were determined based on data obtained from phage annotation (Figure 4). The analysis of the constructed maps showed their similar genomic organization. All the studied genomes contain blocks of genes responsible for genome replication and viral capsid assembly (Figure 4). At the beginning of most genomes, there is a block of the tRNA genes, with the exception of the ctI6W6 phage, in which tRNA genes are located before the genes of structural proteins. Several mobile genetic elements were found in the studied genomes.

Data on the genomic organization, hosts, GC contents, tRNA, suppressor tRNA, DGR, and suppression in various Zeta crAss-like phages (Figure 2, Table 1) indicated that all phages differ significantly from each other.

Two putative genera of Zeta crAss-like phages (clade 8) are most remote from the root of the tree; the subclade 9 contains four isolates, and subclade 10 contains eight phage isolates (Figure 2). All representatives of clade 8 have a suppressor tRNA with a CTA anticodon in their genome, with the exception of the k141_42634 phage genome. However, this MAG has a length of 156.118 bp and might not be a complete genome. There is no suppression in the clade 8 genomes. Notably, the suppressor tRNA cannot be detected in some genomes, despite the obvious codon reassignment. This can be explained either by the incompleteness of the deposited sequences or by the difficulty of identifying the extremely divergent phage-encoded suppressor tRNA.

### 2.3. Comparative Analysis of the Lera Genome with the Similar Genome

The Lera genome exhibits the highest similarity with the ct1yV3 genome. The level of intergenomic similarity between the genomes determined by VIRIDIC is 92.8%, and the level of identity of these sequences is 90.8%. Most of the differences between Lera and ct1yV3 genomes are in their mobile genetic elements (Figure 5). All five ORFs encoding the homing endonuclease of the Lera genome are absent in ct1yV3. In turn, ct1yV3It has three ORFs encoding homing endonuclease, which are absent in the Lera genome. Additionally, there are several differences in the hypothetical proteins, also possibly caused by mobile genetic elements. If we compare the Lera and ct1yV3 genomes without taking into account the difference in the length of the terminal sequences, mobile genetic elements, and genes encoding the ribonucleoside-triphosphate reductase and hypermutated tail fiber protein fragment, the level of identity of their genomes would be 96%. The level of identity of the TerL genes is 99.4%. Apparently, these phages diverged quite recently; however, their inherent mobile elements introduced differences in their genomes. We propose to name Genus 1, shown in Figure 3, which contains the Lera and ct1yV3 Zeta crAss-like phages, the *Echekvirus*.

When comparing the Lera and ct1yV3 phages with the ctE7s22 phage, which forms a common subclade with them (Figure 2), an inversion of the gene block in the ctE7s22 genome can be noted. A comparison of the proteomes of Lera, ctE7s22 phages, and their distant relative C040_19 indicated conserved genes in Zeta crAss-like phages and their similar gene synteny (Figure 5).

### 2.4. Specific Genetic Features of Phages from the Proposed Echekviridae Family

#### 2.4.1. Analysis of the Ribonucleoside-Triphosphate Reductases

An interesting feature of the Lera and ct1yV3 genomes from the proposed *Echekvirus* genus is that they both encode ribonucleoside-triphosphate reductase located in the same place of the genome; however, these proteins are not related. Adenosylcobalamin-dependent ribonucleoside-triphosphate reductase class II is encoded by the Lera genome, whereas ct1yV3 genome encodes anaerobic ribonucleoside-triphosphate reductase class III. The latter enzyme uses an oxygen-sensitive glycyl radical and an iron–sulfur cluster to catalyze the reductive synthesis of deoxyribonucleotides from their corresponding ribonucleotides during anaerobic conditions [37,38]. The ATP-cone domain is located at the N terminus of the protein. Ribonucleoside-triphosphate reductase class III is widely represented in the genomes of Zeta crAss-like phages, while class II enzymes are present in the genomes of only the Lera and ctSlg4 phages (Figure 6).

#### 2.4.2. Analysis of the DNA Polymerases

DNA polymerases of the A and B families have been identified in the genomes of different Zeta crAss-like phages. The B family DNA polymerase was found in the genomes of only six Zeta crAss-like phages, while the family A enzyme was found in the others. A BLASTp search (https://blast.ncbi.nlm.nih.gov/Blast.cgi?PROGRAM=blastp&PAGE_TYPE=BlastSearch&LINK_LOC=blasthome, accessed on 23 May 2025) for proteins similar to them revealed proteins of *Steigviridae* phages similar to DNA polymerase of the family A and proteins of the *Intestiviridae*, *Steigviridae*, *Crevaviridae*, and *Suoliviridae* phages similar to DNA polymerase B. On phylogenetic trees based on DNA polymerases (Figure 7), crAss-like phages do not form monophyletic groups according to the taxonomy. Thus, the ctE7s22 DNA polymerase shows significant similarity to proteins of *Steigviridae* phages (Figure 7A), two proteins of Zeta crAss-like phages are grouped with proteins of *Suoliviridae* phages, and four others with a high degree of reliability form a clade with DNA polymerase of the phi14:2 phage (*Steigviridae*) (Figure 7B).

#### 2.4.3. Analysis of the Diversity-Generating Retroelements (DGRs)

Diversity-Generating Retroelements are unique genetic elements found in bacteria, bacteriophages, and archaea. They are molecular machines that introduce targeted hypermutations into specific genes, generating vast protein diversity. This mechanism is thought to help organisms adapt rapidly to changing environments, evade host immune systems, or diversify ligand-binding capabilities [39,40,41]. DGRs were detected in 18 of the 43 available genomes of Zeta crAss-like phages (Figure 8 and Table 1). Almost all DGRs encode the full-length reverse transcriptases (RTs) and tail fiber proteins (TFPs). The analysis revealed the presence of amber suppression in seven RT genes. This fact makes it difficult to search for DGRs in the genomes. Despite the fact that unlike Epsilon crAss-like phages, in which myDGR program [42] cannot find DGRs due to the presence of the non-standard GxxxSP motif in RT [31], Zeta crAss-like phages contain characteristic GxxxSQ motif found in most RTs in DGRs.

A BLASTp search (https://blast.ncbi.nlm.nih.gov/Blast.cgi?PROGRAM=blastp&PAGE_TYPE=BlastSearch&LINK_LOC=blasthome, accessed on 25 May 2025) for sequences similar to Zeta crAss-like phage RTs was performed. Their significant similarity with proteins of the *Intestiviridae* and *Suoliviridae* phages was found. Phylogeny of RT proteins of 18 Zeta crAss-like phages was analyzed and it was indicated that the Lera and ct1yV3 genomes encode RT, which are substantially different from those of other crAss-like phages (Figure 9). Like DNA polymerases, Zeta crAss-like phages do not form a monophyletic group in the RT proteins’ phylogeny.

When comparing Zeta crAss-like phage RTs, differences in protein sizes can be noted (Figure 8). The RT of the ctSlg4 phage is the shortest (321 aa), whereas the RTs of ctoVY1, cthgQ11, and 3717_85572 phages are the longest (414 aa). RTs of Lera and ct1yV3 phages are 399 aa in length. These differences are due to the high variability of the RTs’ C-termini and do not affect the catalytic domains of proteins. Three-dimensional (3D) structures constructed using AlphaFold3 (Figure 10) indicated that the main differences are located in the C-terminal thumb subdomain, which is responsible for binding RNA template [43]. Given that the RT of the DGR cassette binds to a specific initiation of a mutagenic homing site present in the phage genome, this variability is easy to explain.

A distinctive feature of the Lera phage and relative ct1yV3 (both from the proposed *Echekvirus* genus) is the presence of the accessory genes (AVDs) that make their DGRs complete. In addition, several template repeats (TRs) were found only in the Lera and ct1yV3 genomes (Figure 8). The TR is a non-coding region that is a template for highly mutagenic reverse transcription; it corresponds to the variable repeat (VR)—the target region in the protein coding gene that is diversified by cDNA replacing. Phage RT copies the TR into cDNA, which then replaces the VR, introducing mutations. The RT introduces errors, leading to substitution A for any nucleotide in the resulting cDNA. This creates a hypervariable region in the target gene. In the functioning TR–VR pair, only A mutagenesis is usually observed. As for the Lera and ct1yV3 genomes, only A mutagenesis was indeed found in the TR1–VR1 pair, whereas three other nucleotide substitutions were observed in addition to 12 cases of A mutagenesis in the TR2–VR2 pair, and in the putative TR3–putative VR3 pair, ten other nucleotide substitutions were revealed in addition to 18 cases of A mutagenesis. Possibly, the last two pairs were functioning some time ago and after that, they accumulated random nucleotide substitutions. All this raises questions about the molecular mechanisms of DGR cassettes, which remain unexplored.

Phylogenetic analysis of TFPs encoded by the target genes in various DGRs of Zeta crAss-like phages indicated that they fall into several groups (Appendix A): group 1 (4258_58093, ctZY71, 3486_26535, ct1MR12, ct5Hm1, C017_43), group 2 (ctPYl2, ctoVY1, cthgQ11, 3717_85572), group 3 (ctHMy13), group 4 (ctZYC69, ctH0b1, ctbKI9), group 5 (ctSIg4), group 6 (ctE7s22), and group 7 (Lera and ct1yV3). This grouping is in good agreement with the VIRIDIC clustering (Figure 3). 3D modeling showed that each TFP of Zeta crAss-like phages contains one C-type lectin-like domain at the C-terminus (Figure 11), comprising the variable repeat hypermutated by the DGR system. Proteins of groups 6 and 7 (phages Lera, ct1yV3, and ctE7s22) also include several immunoglobulin (Ig)-like domains, similar to bacterial adhesins (intimin and invasin). Such groups of Ig-like domains are known to provide flexibility and mobility for the molecule [44].

As for the *N*-terminal parts, they looked structured but looked sparse on the models. This indirectly indicates that this part of the protein probably forms a complex with other proteins, which fill the free space. Since most phage receptor proteins such as TFPs and tailspikes form homotrimers, trimer models of the N-terminal parts were also generated. In contrast to the monomeric models, the trimeric models of the N-terminal parts looked plausible, while the prediction confidence (pIDDT index) increased. Most likely, TFPs of Zeta crAss-like phages are homotrimeric proteins with the following structure: an N-terminal stem/fiber followed by three identical C-type lectin domains, responsible for binding (which is characteristic of lectins). This hypothesis is supported by the fact that this part of the protein is subject to randomization under the action of the DGR system. In some cases (proteins of groups 6 and 7), lectin domains were separated from each other by an extended chain consisting of Ig-like domains (Figure 11).

### 2.5. Host Prediction

Host genus prediction for the Lera crAss-like phage and other phages from the proposed *Echekviridae* family was performed using the iPHoP environment with the confidence score cutoff equal to 90. The results of the prediction are given in the Table 1. For the phages with multiple putative hosts, the genera are listed in order of decreasing confidence score. The putative host of the Lera and ct1yV3 phages from the *Echekvirus* genus (clade 5 on Figure 2) are bacteria of the genus *Parabacteroides* belonging to the family *Tannerellaceae*, (order *Bacteroidales*, class *Bacteroidia*, phylum *Bacteroidetes*). Members of the genera *Bacteroides*, *Prevotella,* and *Phocaeicola* of the order *Bacteroidales* were the possible hosts for most of the phages under consideration. The genus *Lachnospira* from the family *Lachnospiraceae* (order *Clostridiales*, class *Clostridia*, phylum *Firmicutes*) was also frequently encountered. It is worth noting that all of the above bacteria are part of the resident human gut microbiota. In some cases, either *Lachnospira* or *Bacteroides* belonging to different phyla were identified as hosts with similar reliability (Table 1).

## 3. Discussion

CrAss-like phages are the most common components of mammalian intestinal viromes. However, until now, their life cycle, functioning, evolution, and mechanisms of penetration into the cell and counteraction to antiphage defense systems remain poorly understood [4,6,13,31,45]. In this study, the complete genome of a novel phage named Lera was identified in the virome of a healthy person. Approximately 18% of all ORFs contained suppressor TGA stop codons, and most parts of these genes are located in the block of the essential structural protein genes in addition to the TerL gene. Both tRNA-SUP-CTA, and tRNA-SUP-TCA genes were identified in the Lera phage genome, as well as genes of proteins involved in tRNA metabolism. A detailed comparative analysis of the Lera genome and genomes of relative phages indicated that this phage belongs to the Zeta crAss-like phage clade.

The study revealed significant diversity among Zeta crAss-like phages, which makes it possible to identify a lot of separate genera within this proposed family. This conclusion is supported by a study of the phage proteomes and phylogenetic analysis of key proteins such as the large terminase subunit and RNA polymerase. We separated the studied Lera and closely related ct1yV3 phages into the putative *Echekvirus* genus.

A previous study highlighted that Zeta clade phages contain numerous self-splicing introns and inteins [4]. This trend was also observed in the studied phages, including Lera and ct1yV3 phages from the *Echekvirus* genus, which exhibited high sequence homology with the exception of the different mobile genetic elements. The prevalence of these elements raises questions about their functional role and the ability of phages to maintain efficient replication despite their presence.

The genetic distances between the genomes of Zeta crAss-like phages are very high, which indicates a high rate of evolution of these phages. Mobile genetic elements are absent in the Epsilon crAss-like phages genomes, as well as in Alpha-Gamma crAss-like phage genomes [4]; they are rarely found in the genomes of Delta phages, in contrast to the highest representation of such elements in Zeta crAss-like phage genomes. It can be assumed that mobile genetic elements spur the evolution of Zeta crAss-like phages. The differences in the evolution of Zeta and Epsilon crAss-like phages are particularly pronounced. While Epsilon crAss-like phages are divided only into two putative genera, *Epsilonunovirus* and *Epsilonduovirus* [31], and show high similarity of their genomes within these genera, Zeta crAss-like phages exhibit substantial diversity of their genomes.

CrAss-like phages primarily target Gram-negative, non-spore-forming bacteria of the *Bacteroidales* order, particularly *Bacteroides* species, which are abundant in the gut microbiota of humans and other warm-blooded animals [1,2,3,46,47]. These bacteria play critical roles in host metabolism, such as bile acid deconjugation and the breakdown of proteins and complex sugars [48,49]. Characterization of the Zeta crAss-like phages’ host range revealed a wide variety of potential hosts of these phages. Many phages infect various *Bacteroidetes*; however, this phage family can include viruses infecting bacteria from other bacterial phyla, such as *Firmicutes* and *Bacillota*. The high rate of evolution of Zeta crAss-like phages and the presence of protein diversification machines may contribute to the emergence of virus variants infecting new hosts.

Notably, *Bacteroides* possess unique O-glycosylation systems and can produce diverse capsular polysaccharides to evade phage attacks [50,51]. To counteract these defenses, crAss-like phages may employ diversity-generating retroelements (DGRs) to create hypervariable tail proteins. DGRs are fascinating genetic elements that drive rapid protein diversification, offering evolutionary advantages to viruses. When studying DGR cassettes, the question arises about their operability. Are these systems still in operation, or are we seeing the results of their work in the past? In this regard, it can be noted that all the studied DGR cassettes of Zeta crAss-like phages contain full-length genes encoding RT (Figure 8), whereas the Alpha and Epsilon crAss-like phages contain both complete DGR cassettes and their fragments [31], indicating DGR diversification of both silent and functional cassettes. Apparently, this mechanism is still relevant for the studied Zeta crAss-like phages.

Notably, TFPs encoded by the target genes in various DGRs of Zeta crAss-like phages are diverse in their architecture (Figure 11). The sequences of the N-terminal part, which is presumably involved in docking with other structural proteins of the virion, also differ. However, these proteins have several similar properties: the presence of a C-type lectin domain at the C-terminus and their probable ability to form homotrimers. TFPs of groups 6 and 7 (the proposed *Echekvirus* genus) contain several Ig-like domains at the C-terminus, similar to bacterial adhesives (Figure 11). It can be assumed that the presence of a flexible spacer at the C-terminus of the protein facilitates scanning of the host cell surface and allows the lectin domains to attach to receptors distributed across the cell surface.

The reason for the diversity of TFPs in Zeta crAss-like phages is unclear. This may indicate both the use of different hosts by different Zeta crAss-like phages and the ability to adapt to changing receptors on the surface of the host cell. As a whole, this testifies that such phages can recognize a variety of different structures.

Both phages from the putative *Echekvirus* genus (Lera and ct1yV3) have unique feature in their DGRs. Only Lera and ct1yV3 from Zeta crAss-like phages contain the AVD protein genes (Figure 8), and their RT differs significantly from the RT of other Zeta crAss-like phages (Figure 9). In addition, Lera and ct1yV3 have three TR–VR pairs (Figure 8), which differ among themselves in the number of non-canonical mutagenesis cases for DGR cassettes.

A notable peculiarity of Zeta crAss-like phages is the presence of suppressor tRNAs, which complicate genome annotation by fragmenting genes into short ORFs when standard genetic code is applied. Understanding the diversity of phages and conducting thorough virome analyses is challenging without considering stop-codon recoding. Zeta crAss-like phages exhibit diverse patterns of stop-codon suppression, utilizing either TGA (opal) or TAG (amber) codons. Some of these phages carry suppressor tRNAs that appear inactive, as the corresponding stop codons are either absent or followed shortly by a TAA (ochre) stop codon. In other cases, either TGA or TAG codons are embedded within essential viral genes, such as those encoding structural proteins or DNA-processing enzymes. For example, the Lera phage possesses both TGA and TAG suppressor tRNAs but employs only one, raising questions about their functional role.

Suppressor tRNAs have been identified in various phages, including phAss-1, Black, Sapphire, Jade, and Agate phages [52,53,54,55,56,57,58]. While some studies propose that phage hosts may also use alternative genetic codes, evidence suggests that many hosts adhere to the standard code [33,59]. Instead, stop codon recoding may serve as a post-transcriptional regulatory mechanism, particularly for late-stage genes involved in virion assembly and host cell lysis [32,33]. In temperate phages, this mechanism might also facilitate the switch from lysogenic to lytic cycles [32].

In the case of Zeta crAss-like phages, we are apparently dealing with a lytic phage. Why do such phages need “non-functioning” suppressor tRNAs? One hypothesis is that the presence of a suppressor tRNA in the absence of apparent codon reassignment represents an early stage of genetic code evolution, and the genomes of such phages have not yet had time to accumulate suppressed stop codons in the conserved genes [4]. However, such an assumption seems unlikely. For some Zeta crAss-like phages, it can be observed that phages which have significantly diverged from each other continue to maintain the same strategy of using suppressor tRNAs and do not accumulate stop codons in the genome. What is the benefit of such a strategy? It can be hypothesized that suppressor tRNAs are necessary for phages to counteract bacterial defense systems, leading to a decrease in the production of host antiphage proteins. In fact, it has been shown that the production of suppressor tRNA in *Escherichia coli* leads to a change in the expression of individual genes [60,61]. It has been previously revealed that suppression of the amber codon in *E. coli* by plasmids causes global proteomic changes with a marked increase in the production of the YdiI protein, for which the function of expelling plasmids from the cell was identified [61]. Another study has shown that the suppression of amber stop codons reduces the pathogenicity of *Salmonella* spp. [62].

The genetic code in bacteria has been found to include numerous variations, such as codon reassignment and recoding. In some bacteria, UGA codons can be recoded both as selenocysteine and stop codons. This recoding refers to site-specific codon reassignment that depends on the mRNA sequence context. Another well-known example is the reassignment of the UAG stop code to pyrrolysine [63]. Bacteria with recoded genomes have evolved over extended periods, fine-tuning their translation machinery for optimal performance. However, the introduction of foreign suppressor tRNAs can be toxic for bacterial cells [64,65]. Thus, it is possible that the massive production of suppressor tRNAs weakens the host and is beneficial to the phage; two different suppressor tRNAs should enhance this effect, though their exact function requires further experimental investigation.

In the case of the *Echekvirus* genus (Lera and ct1yV3 phages), opal suppression of signature genes and the presence of suppressor tRNAs for both TGA and TAG stop codons are observed. However, their closest phage, ctE7s22, has no signs of genome recoding, despite there being a suppressor tRNA of the TAG stop codon. Apparently, the opal suppression system in the Lera and ct1yV3 phages developed after divergence from a common ancestor with ctE7s22. At the same time, the TAG stop codon in the genome of these three phages is rare, and in all cases, another stop codon is located shortly after it. So, such an organization of genomes appeared during a long evolution.

A distinctive feature of Zeta crAss-like phages is the frequent switching of DNA polymerase types. It has been suggested that these switches may serve as an evasion strategy against bacterial defense mechanisms, particularly those targeting phage DNA replication [4,66]. The widespread occurrence of such in situ DNA polymerase replacements highlights the evolutionary tactics crAss-like phages employ to overcome host resistance directed to phage DNA polymerase and maintain their reproductive efficiency. Proceeding from the fact that the phylogeny based on DNA polymerase A and B families does not group phages according to the similarity of their genomes and their taxonomy, it can be concluded that there is a frequent exchange of genes of these proteins not only between A and B families, but also within families. Perhaps there is an unknown mechanism contributing to frequent DNA polymerase replacements.

## 4. Materials and Methods

### 4.1. Virome Sequencing and Analysis

Viral DNA was extracted from a fecal sample following a previously published protocol [52]. In short, the sample was clarified through multiple centrifugation steps and then treated with DNase I and Proteinase K (Thermo Fisher Scientific, Waltham, MA, USA). The DNA was purified using phenol-chloroform extraction followed by ethanol precipitation. The purified DNA was resuspended in 50 µL of TE buffer, and its concentration was quantified with a Qubit 4.0 fluorometer (Thermo Fisher Scientific, Waltham, MA, USA). A virome shotgun library was prepared using the NEB Next Ultra DNA Library Prep Kit (New England Biolabs, Ipswich, MA, USA). Sequencing was carried out on an Illumina MiSeq sequencer with a MiSeq Reagent Kit 2 × 250 v.2 (Illumina Inc., San Diego, CA, USA). The reads were assembled into contigs using SPAdes v.3.15.2 [67]; putative phage sequences were identified with VIBRANT v.1.2.1 [68]. One of the resulting MAGs was subjected to further analysis. This work was approved by the Local Ethics Committee of the Center for Personalized Medicine, Novosibirsk (protocol #2, 12 February 2019). The written consent of the healthy volunteer was obtained in accordance to the guidelines of the Helsinki ethics committee.

### 4.2. In Silico Host Prediction

For the identified potential phage sequence, bacterial host prediction was performed using the integrated machine learning environment iPHoP v.1.4.1 [69], which combines various tools used for this task: similarity search between the virus genome and the host genome, CRISPR analysis, k-mer frequency analysis, and comparison with a database of already studied bacteriophages with known hosts.

### 4.3. Phage Genome Analysis

PhageTerm v.1.0.12 software [70] (https://galaxy.pasteur.fr; accessed on 20 May 2025) was applied to find the position of the phage’s termini. VectorBuilder’s GC Content Calculator (https://en.vectorbuilder.com/tool/gc-content-calculator.html, accessed on 24 May 2025) was used to determine the GC content of the phage. To analyze genomes, the Ori-Finder 2022 software was used [71] (https://tubic.org/Ori-Finder, accessed on 24 May 2025). Phage genomes were searched for tRNA genes using tRNAscan-SE v.2.0 software [72] (http://trna.ucsc.edu/tRNAscan-SE/, accessed on 1 June 2025).

Putative open reading frames (ORFs) were determined and annotated using Rapid Annotation Subsystem Technology (RAST) v.2.0 [73] (https://rast.nmpdr.org, accessed on 12 June 2025). The identified ORFs were verified manually using BLAST algorithms against nucleotide and protein sequences, deposited in the NCBI GenBank (https://ncbi.nlm.nih.gov, accessed on 2 June 2025). In addition, the InterProScan (https://www.ebi.ac.uk/interpro/search/sequence/, access date 25 June 2025), HHPred (toolkit.tuebingen.mpg.de/tools/hhpred, access date 23 June 2025), and HHblits (toolkit.tuebing-en.mpg.de/tools/hhblits, access date 23 June 2025) were applied for the identification of hypothetical proteins. To edit and align the nucleotide sequences, BioEdit v.7.2.5 [74] and MAFFT v.7 (https://mafft.cbrc.jp/alignment/server, accessed on 30 June 2025) tools were used.

Phylogenetic trees based on maximum likelihood were constructed using IQ-tree software version 2.0.5 [75], employing the optimal substitution model identified by ModelFinder [76]. Branch support was evaluated with 500 bootstrap iterations. The final trees were midpoint-rooted and graphically represented using FigTree version 1.4.1.

A comparative study of the Lera phage proteome was conducted using the ViPTree version 3.7 web server (available at https://www.genome.jp/viptree, accessed on 1 June 2025) with default parameters [77]. ViPTree analysis that relied on genome-wide sequence comparisons was carried out using the tBLASTx algorithm. Intergenomic similarities were assessed with the VIRIDIC v1.1 tool (accessible at http://rhea.icbm.uni-oldenburg.de/VIRIDIC, accessed on 5 June 2025) [78]. Genera predictions were made based on a 70% intergenomic identity threshold [35,36].

### 4.4. 3D Modeling of Protein Structure

The structure of proteins was predicted using AlphaFold3 [79] (https://alphafoldserver.com/welcome, accessed on 2 July 2025). The backbone conformation was considered reliably predicted for regions with a pLDTT value of 70% or greater. The model of the complete protein was built by combining overlapping high confidence models of fragments if some of its extended regions were predicted with low confidence. Model building and visualization were performed in UCSF Chimera v 1.13.1 [80]. The analysis of domain structural similarity was performed using the Dali web server v.5 [81] (http://ekhidna2.biocenter.helsinki.fi/dali/, accessed on 3 July 2025).

## 5. Conclusions

Zeta crAss-like phages have a high diversity of genomes and exhibit complex genomic features, including mobile genetic elements, DGR cassettes, suppressor tRNAs, and dynamic DNA polymerases, which may enhance their adaptability and evasion of host defenses. Currently, Zeta crAss-like phages are presented in databases without taxonomic classification. For the taxonomic binning of new sequences and for effective further study of Zeta crAss-like phages, which is necessary to fully understand their evolutionary and functional significance, it is necessary to classify them. Following the comprehensive study, we suggest assigning the designation *Echekviridae* to the Zeta crAss-like phage family.

## Figures and Tables

**Figure 1 ijms-26-07694-f001:**
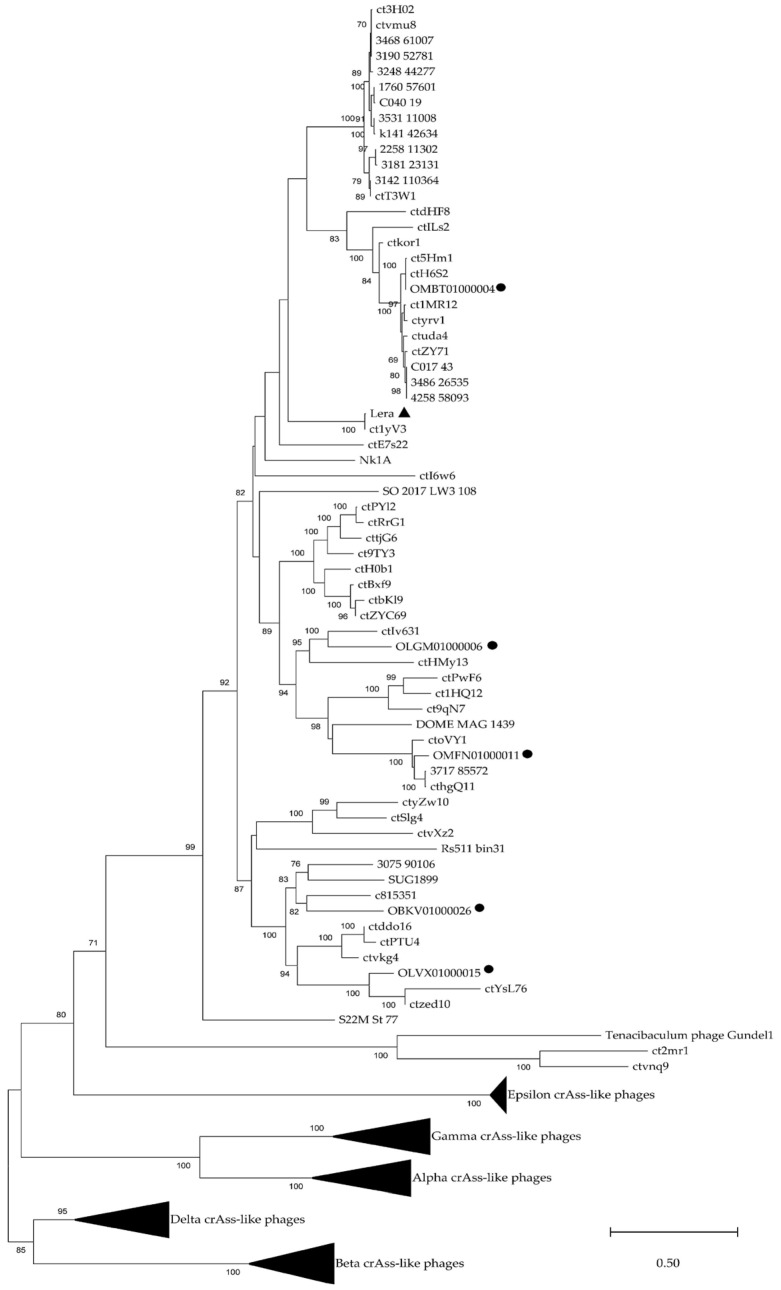
Maximum likelihood phylogenetic tree of the terminase large subunit generated using IQ-tree software. The investigated sequence of Lera is marked with a black triangle. Sequences from study [4] are marked with black circles. Bootstrap values calculated from 500 replicates are given at the nodes. The scale bar represents the number of substitutions per site.

**Figure 2 ijms-26-07694-f002:**
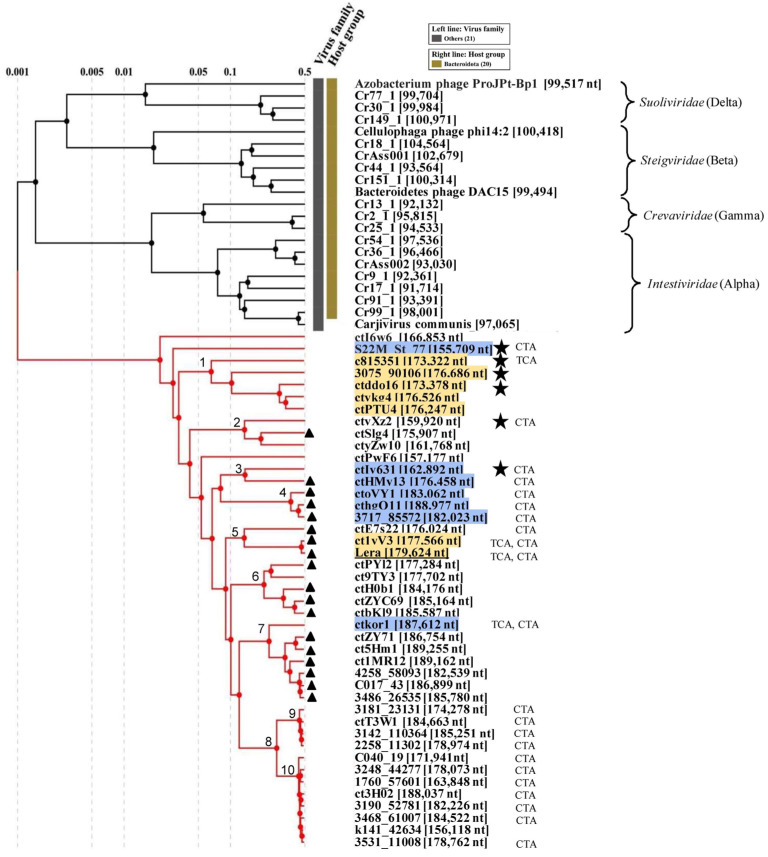
ViPTree analysis of the Lera phage (underlined). Dendrogram plotted by ViPTree version 3.7 using Lera and phages with SG values > 0.001. Phage sequences downloaded manually from NCBI GenBank are marked with red phylogenetic branches. Phages with opal and amber suppression in their genomes are highlighted in yellow and blue, respectively. Genomes with DGRs are marked with black triangles. Phages with the polB gene in the genome are marked with black asterisks. Anticodon(s) of the suppressor tRNAs are shown at the right. The nodes of the tree contain the numbers of clades discussed in the text.

**Figure 3 ijms-26-07694-f003:**
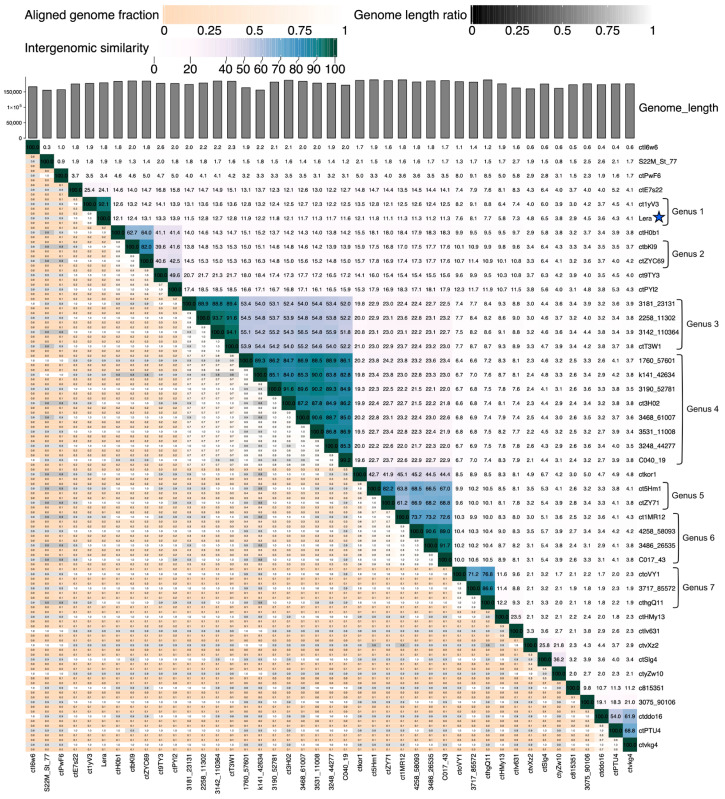
Bidirectional clustering heatmap visualizing a VIRIDIC-generated similarity matrix for Lera and related phage genomes. The Lera phage is marked with a blue asterisk.

**Figure 4 ijms-26-07694-f004:**
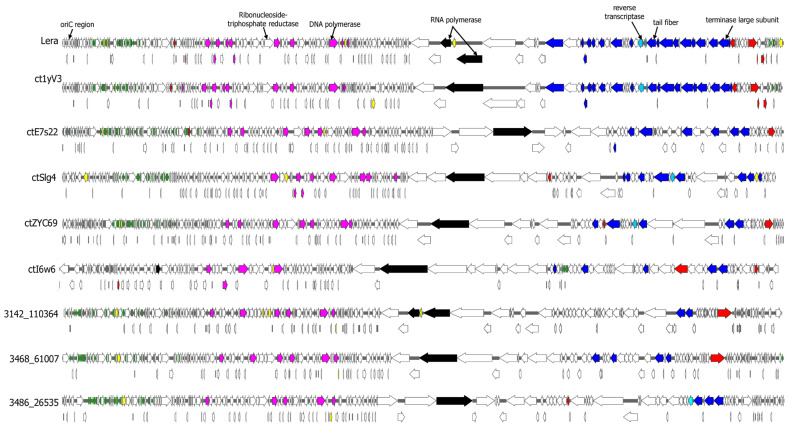
The whole genome maps of Lera phage and 8 other Zeta crAss-like phages. ORFs are colored according to their proposed function: DNA replication—pink; head assembly and structural—blue; transcription—black; reverse transcriptase and AVD genes—sky blue; lysine genes—red; mobile genetic elements—yellow; tRNA—green.

**Figure 5 ijms-26-07694-f005:**
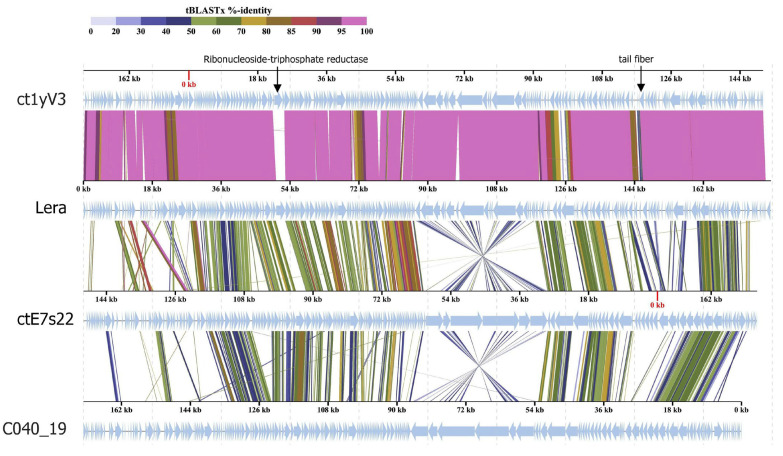
Comparative genome alignment of the Lera and genomes of ct1yV3, ctE7s22, and C040_19 phages. Analysis was performed using VipTree software v. 3.7. The percentage of sequence similarity is indicated in color; the color scale is shown at the top.

**Figure 6 ijms-26-07694-f006:**
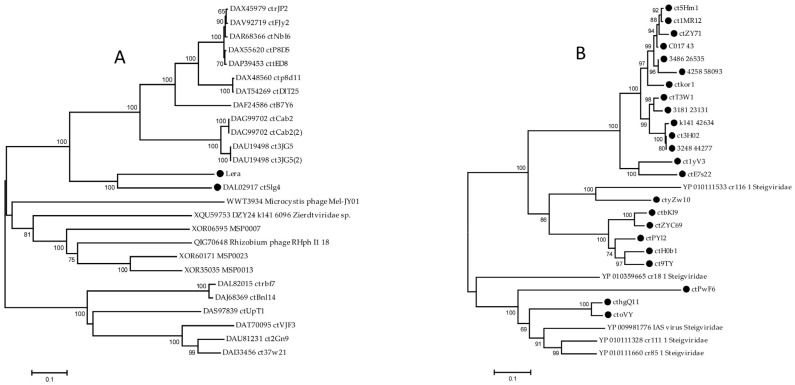
Maximum likelihood phylogenetic tree of the class II ribonucleotide reductase (**A**) and class III ribonucleotide reductase (**B**) from the Zeta crAss-like phages generated using IQ-tree software v. 2.0.5. The sequences of Zeta crAss-like phage are marked with black circles. Bootstrap values calculated from 500 replicates are given at the nodes. The scale bar represents the number of substitutions per site.

**Figure 7 ijms-26-07694-f007:**
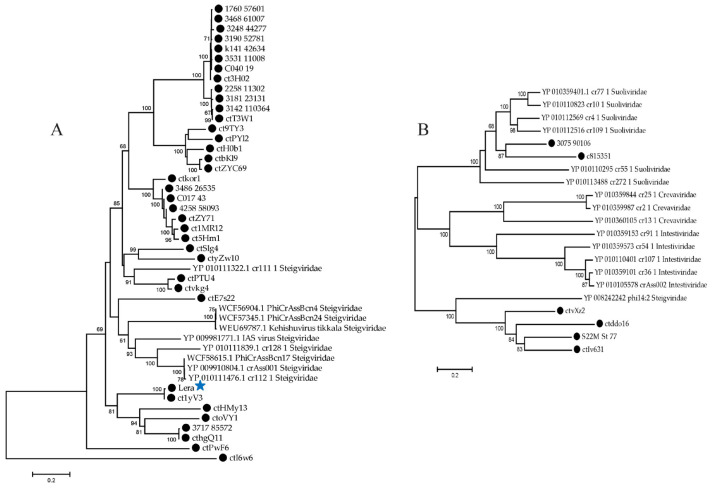
Maximum likelihood phylogenetic tree of the Zeta crAss-like phage DNA polymerase of family A (**A**) and family B (**B**) generated using IQ-tree software. The sequences of Zeta crAss-like phage are marked with black circles. Bootstrap values calculated from 500 replicates are given at the nodes. The scale bar represents the number of substitutions per site. The Lera phage is marked with a blue asterisk.

**Figure 8 ijms-26-07694-f008:**
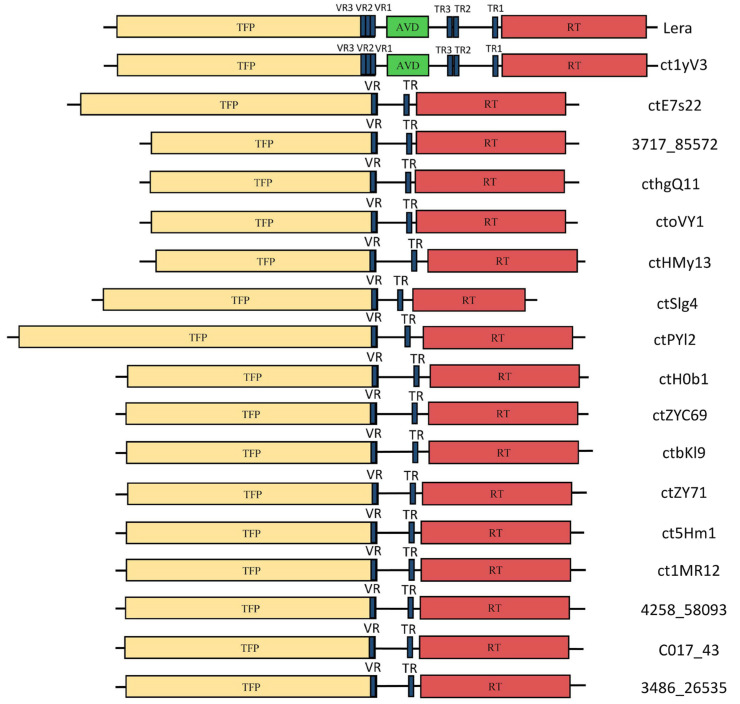
DGR cassettes found in the available genomes of Zeta crAss-like phages. TFP—tail fiber protein gene, VR—variable region, AVD—accessory gene, TR—template repeat, RT—reverse transcriptase gene.

**Figure 9 ijms-26-07694-f009:**
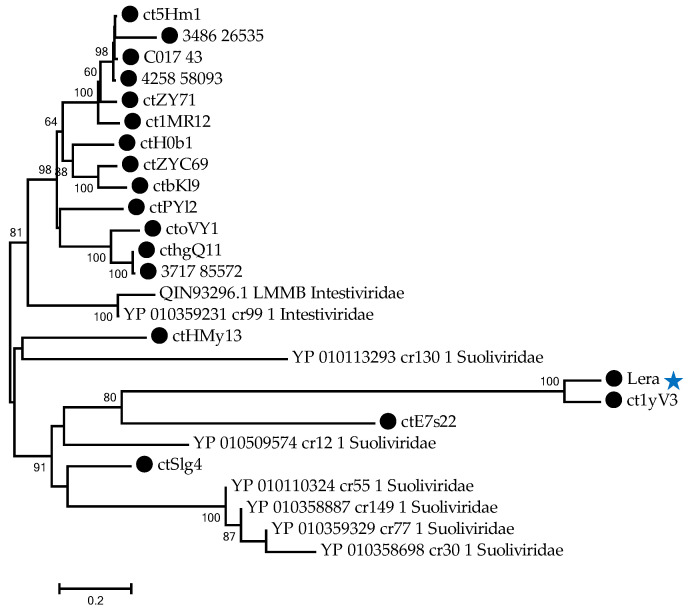
Maximum likelihood phylogenetic tree of the reverse transcriptase (RT) generated using IQ-tree software. RT sequences of Zeta crAss-like phages are marked with black circles. Bootstrap values calculated from 500 replicates are given at the nodes. The scale bar represents the number of substitutions per site. The sequence from the Lera phage is marked with a blue asterisk.

**Figure 10 ijms-26-07694-f010:**
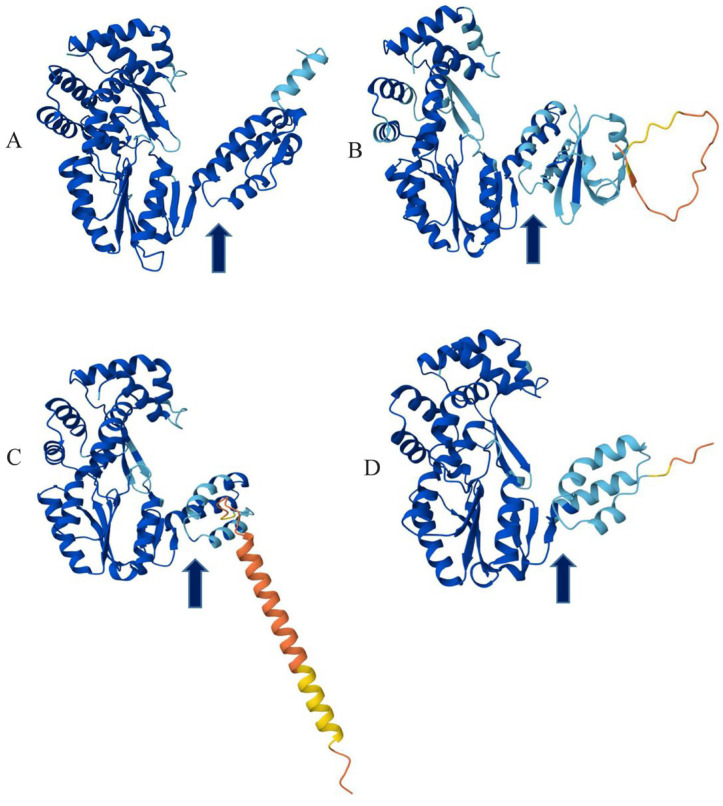
Ribbon representation of RT 3D structures encoded by the phages Lera (**A**), ctoVY1 (**B**), 3486_26535 (**C**), and ctSlg4 (**D**). The C-terminal thumb subdomains are marked with arrows. 3D models were predicted using AlphaFold3. Blue indicates very high confidence, sky blue indicates high confidence, yellow indicates low confidence, and red indicates very low confidence of the model.

**Figure 11 ijms-26-07694-f011:**
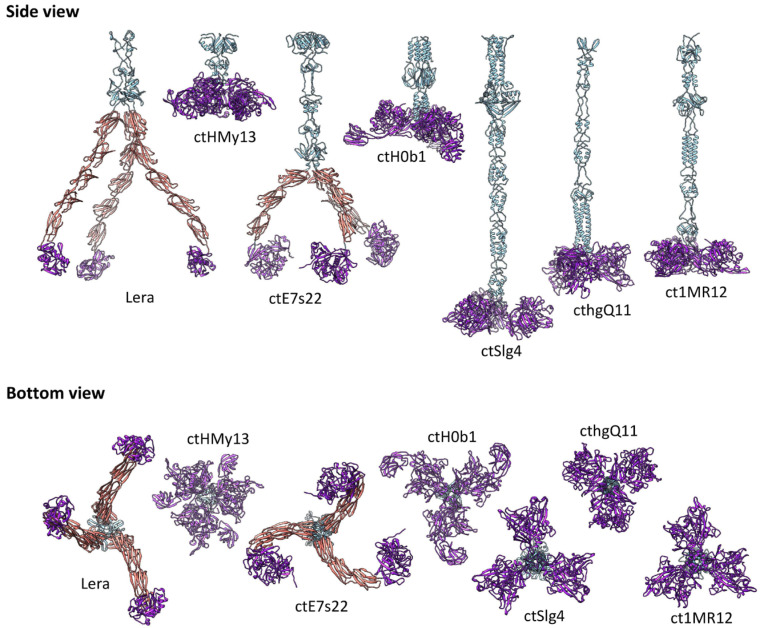
3D models of tail fiber proteins (TFPs) encoded by the target genes in the DGR cassettes. Trimeric fiber regions are shown in sky blue; C-type lectin domains containing VR-encoded region are shown in purple. Ig-like domains are shown in pink. Initial models were generated using Alphafold3; final models were built and visualized using UCSF Chimera v.1.13.

**Table 1 ijms-26-07694-t001:** Characteristics of the available Zeta crAss-like phage genomes.

##	Name of Phage	Genome Length	# of tRNA	Putative Host	DNA Polymerase	GC Content, %	SG *	Anticodon(s) Suppressor tRNA **	Suppression	DGR
1.	ctI6w6	166,853	6	*Prevotella*	polA	37.61	0.0671	-	-	-
2.	S22M_St_77	155,709	17	-	polB	35.72	0.0648	**CTA**	amber	-
3.	c815351	173,322	23	-	polB	33.17	0.0860	**TCA**	opal	-
4.	3075_90106	176,686	25	*Holdemanella*	polB	30.50	0.0845	-	opal	-
5.	ctddo16	173,378	23	-	polB	32.33	0.0781	-	opal	-
6.	ctvkg4	176,526	25	*Faecalibacillus*/*Holdemanella*	polA	33.16	0.0849	-	opal	-
7.	ctPTU4	176,247	18	*Faecalibacillus*	polA	32.42	0.0867	-	opal	-
8.	ctvXz2	159,920	6	-	polB	28.04	0.1011	-	-	-
9.	ctSlg4	175,907	28	*Butyricimonas*	polA	29.31	0.1206	-	-	+
10.	ctyZw10	161,768	6	*Bacteroides*/*Gabonibacter*	polA	33.56	0.1027	-	-	-
11.	ctPwF6	157,177	4	-	polA	30.45	0.0915	-	-	-
12.	ctIv631	162,892	21	-	polB	32.73	0.1240	**CTA**	amber	-
13.	ctHMy13	176,458	23	*Prevotella*	polA	35.10	0.1194	**CTA**	amber	+
14.	ctoVY1	183,062	27	*Phocaeicola*/*Lachnospira*	polA	34.75	0.1259	**CTA**	amber	+
15.	cthgQ11	188,977	27	*Prevotella*/*Lachnospira*	polA	34.65	0.1375	**CTA**	amber	+
16.	3717_85572	182,023	24	*Lachnospira*/*Prevotella*	polA	34.46	0.1383	**CTA**	amber	+
17.	ctE7s22	176,024	31	-	polA	31.87	0.2762	CTA	-	+
18.	ct1yV3	177,566	25	*Parabacteroides*	polA	32.36	0.9323	**TCA**, CTA	opal	+
19.	**Lera**	**179,624**	**26**	** *Parabacteroides* **	**polA**	**32.36**	**1**	**TCA, CTA**	**opal**	**+**
20.	ctPYl2	177,284	21	*Phocaeicola*/*Bacteroides*	polA	33.41	0.1916	-	-	+
21.	ct9TY3	177,702	19	*Bacteroides*	polA	33.49	0.1821	-	-	-
22.	ctH0b1	184,176	26	-	polA	34.30	0.1822	-	-	+
23.	ctZYC69	185,164	25	*Bacteroides*	polA	34.32	0.1841	-	-	+
24.	ctbKl9	185,587	23	*Bacteroides*	polA	34.57	0.1849	-	-	+
25.	ctkor1	187,612	32	*Bacteroides*/*Lachnospira*	polA	31.36	0.1578	**CTA**, TCA	amber	-
26.	ctZY71	186,754	34	*Lachnospira*	polA	36.46	0.1677	-	-	+
27.	ct5Hm1	189,255	32	*Lachnospira*	polA	36.54	0.1774	-	-	+
28.	ct1MR12	189,162	29	*Bacteroides*/*Phocaeicola*	polA	36.40	0.1733	-	-	+
29.	4258_58093	182,539	31	*Lachnospira*	polA	36.56	0.1696	-	-	+
30.	C017_43	186,899	34	*Bacteroides*/*Lachnospira*	polA	36.50	0.1714	-	-	+
31.	3486_26535	185,780	31	*Bacteroides*/*Phocaeicola*	polA	36.49	0.1724	-	-	+
32.	3181_23131	174,278	25	*Bacteroides*	polA	35.35	0.1825	CTA	-	-
33.	ctT3W1	184,663	26	*Bacteroides*	polA	35.41	0.1914	CTA	-	-
34.	3142_110364	185,251	23	*Bacteroides*	polA	35.42	0.1917	CTA	-	-
35.	2258_11302	178,974	20	*Bacteroides*	polA	35.38	0.1893	CTA	-	-
36.	C040_19	171,941	23	*Bacteroides*	polA	36.50	0.1812	CTA	-	-
37.	3248_44277	178,073	29	*Bacteroides*/*Phocaeicola*	polA	36.78	0.1776	CTA	-	-
38.	1760_57601	163,848	27	*Bacteroides*	polA	36.88	0.1880	CTA	-	-
39.	ct3H02	188,037	29	*Bacteroides*	polA	36.78	0.1838	CTA	-	-
40.	3190_52781	182,226	28	*Bacteroides*	polA	36.74	0.1740	CTA	-	-
41.	3468_61007	184,522	29	*Bacteroides*	polA	36.81	0.1762	CTA	-	-
42.	k141_42634	156,118	17	*Bacteroides*/*Lachnospira*	polA	36.89	0.2029	-	-	-
43.	3531_11008	178,762	28	*Bacteroides*	polA	36.84	0.1776	CTA	-	-

* SG values were calculated according to Bhunchoth et al. [34] as normalized tBLASTx scores (SG; 0 ≤ SG ≤ 1) compared to Lera. ** The anticodons of suppressor tRNAs used by the phage to suppress stop codons are highlighted in bold.

## Data Availability

The complete genome sequence of Lera phage was submitted into the GenBank database with the accession number PV725988.

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
