# Peer review of "Zeta CrAss-like Phages, a Separate Phage Family Using a Variety of Adaptive Mechanisms to Persist in Their Hosts"

_ijms, 2025, doi:10.3390/ijms26167694_

Round 1

Reviewer 1 Report

Comments and Suggestions for Authors

The Results section presents a series of excellent, but somewhat disconnected, analyses. The manuscript would benefit from stronger "signposting" to guide the reader through the logical progression of the investigation. For instance, after establishing the overall taxonomy with VIRIDIC and ViPTree, the authors could more explicitly state that they will now delve into the specific genetic features (DGRs, polymerases, etc.) that drive this observed diversity. This would help weave the individual analyses into a more cohesive and compelling story.

The Discussion section could be made more impactful by slightly expanding on the functional implications of the observed phenomena. For example, the presence of inactive suppressor tRNAs is a fascinating observation. The authors hypothesize this may be a strategy to weaken the host. This is an intriguing idea that could be strengthened by citing any additional literature on how foreign tRNAs can disrupt host translation or by more explicitly framing it as a key hypothesis to be tested in future experimental studies.

There appears to be a duplication error in the Methods Section. Section 4.2 and Section 4.5 are both titled "In Silico Host Prediction" and contain identical text.

Several web tools in the Methods section are cited with "accessed on" dates in the future (e.g., June 2025). This should be corrected to reflect the actual dates of access.

Figure 2 Legend is very detailed and helpful. A minor clarification could be to explicitly state what the "red stars" signify, as "Phage sequences downloaded manually..." is also used to describe the red branches.

For Figure 11, the manuscript could briefly mention the potential significance of the flexible Ig-like domains in phages Lera, ct1yV3, and ctE7s22—perhaps that they act as rigid spacers, allowing the receptor-binding domains greater reach or flexibility to scan the host cell surface. This is alluded to in the text but could be reinforced in the context of the figure.

Author Response

Reply to reviewer's comments

We sincerely appreciate the reviewer's ideas and suggestions, which helped us to refine and improve the revised version of our manuscript.

  1. The Results section presents a series of excellent, but somewhat disconnected, analyses. The manuscript would benefit from stronger "signposting" to guide the reader through the logical progression of the investigation. For instance, after establishing the overall taxonomy with VIRIDIC and ViPTree, the authors could more explicitly state that they will now delve into the specific genetic features (DGRs, polymerases, etc.) that drive this observed diversity. This would help weave the individual analyses into a more cohesive and compelling story.
    Response: Thank you for the idea to make the manuscript clearer. We have additionally divided the Results into the sections and subsections suggested by the reviewer. See "Results".
  1. The Discussion section could be made more impactful by slightly expanding on the functional implications of the observed phenomena. For example, the presence of inactive suppressor tRNAs is a fascinating observation. The authors hypothesize this may be a strategy to weaken the host. This is an intriguing idea that could be strengthened by citing any additional literature on how foreign tRNAs can disrupt host translation or by more explicitly framing it as a key hypothesis to be tested in future experimental studies.
    Response: We appreciate the reviewer's recommendation and have expanded this paragraph in the revised manuscript. We have added additional citations to the revised text. Lines 497-507.
  1. There appears to be a duplication error in the Methods Section. Section 4.2 and Section 4.5 are both titled "In Silico Host Prediction" and contain identical text.
    Response: The text has been corrected.
  1. Several web tools in the Methods section are cited with "accessed on" dates in the future (e.g., June 2025). This should be corrected to reflect the actual dates of access.
    Response: The text has been corrected.
  1. Figure 2 Legend is very detailed and helpful. A minor clarification could be to explicitly state what the "red stars" signify, as "Phage sequences downloaded manually..." is also used to describe the red branches.
    Response: Figure 2 and Legend has been corrected.
  1. For Figure 11, the manuscript could briefly mention the potential significance of the flexible Ig-like domains in phages Lera, ct1yV3, and ctE7s22—perhaps that they act as rigid spacers, allowing the receptor-binding domains greater reach or flexibility to scan the host cell surface. This is alluded to in the text but could be reinforced in the context of the figure.
    Response: We appreciate your suggestion. We have expanded the Discussion to include a consideration of the potential significance of the flexible spacer in the TFPs of these phages.  Lines 447-451.

Reviewer 2 Report

Comments and Suggestions for Authors

Babkin et al. present a very interesting, to the point, and well written manuscript providing compelling support for classifying a group of Zeta crAss-like phages into a separate phage family. The author propose that the Family name should be Echekviridae to signify the source, being "intestine".

The authors used extensive in silico meta genomic analyses to assemble sequences from a virome shotgun library constructed from a stool sample.  Ultimately, to analyze Zeta crAss-like genomes, 42 sequences (each >150 kb) were chosen from the database, based on the TerL gene related to Lera phage identified initially in the stool sample study.

Further analysis revealed that Lera and closely associated phages have unique features that validate their classification as a separate clade and a new family Echekviridae.

Some of these special features of the proposed Echekviridae family when compared to crAss-like phage families are as follow:

They exhibit greater genome variability, use alternative genetic coding, and interestingly contain unique diversity-generating retroelements (DRGs) that mutate tail fiber proteins, presumable to enhance host adaptability to changing host receptors (and perhaps host range including Firmicutes and Bacillota?). These phages also code for unique tail fiber proteins not commonly found in crAss-like phages, and harbor distinct DRG cassette structure, including rare Avd protein genes and TR-VR pair. In addition, they appear to have high rates of genetic diversification that are driven by mobile genetic elements (introns/inteins).

The Reviewer agrees with the authors that the in in silico/metagenomic analyses provide very strong support and justification for proposing a new family of the Zeta crAss-like phages, i.e., Echekviridae. (Hopefully, the ICTV will agree with the proposal.)

Author Response

Reply to reviewer's comments

Response: We would like to thank the reviewer for his/her positive feedback on our work. We appreciate the time and effort spent in evaluating our article.